# TimeGEN: A Cross-Domain and Generative Model for Time Series Forecasting

## Abstract

We propose `TimeGEN`, a lightweight, MLP-based generative deep learning architecture for Transfer Learning in time series forecasting. We use a variational encoder to capture high-level temporal representations across diverse series and domains. To further strengthen this generalization, we combine a reconstruction and forecasting loss, which shapes the latent space to retain local detail while capturing global predictive dependencies. In addition, temporal normalization ensures robustness to varying input scales and noise. To capture multiscale dynamics, we integrate a modular decoder that combines neural basis expansion with multi-rate interpolation, balancing long-range trends with high-frequency variations. Extensive empirical results across ten public datasets demonstrate that `TimeGEN` consistently outperforms SOTA methods in zero-shot and cross-domain settings. In cross-domain settings, `TimeGEN` reduces forecasting error by more than 8% and up to 38%, while achieving a 2–30× speedup in training time compared to SOTA MLP and Transformer methods.

## 1 Introduction

Time series forecasting plays a vital role in decision-making across domains such as energy dispatch, traffic control, supply chain management, and financial planning. While deep learning has led to substantial improvements in forecasting accuracy, most models remain highly sensitive to distributional shifts. Variations in sampling frequency, forecast horizon, covariates, or domain-specific properties, e.g., scale, seasonality, sparsity, or volatility, can significantly degrade performance. As a result, models typically require tuning or retraining when applied to new domains. Moreover, in dynamic environments, models may face concept drift, where the underlying data distribution changes over time. These issues limit their ability to generalize and remain accurate, highlighting the need for transfer learning (TL) and adaptive approaches that can leverage prior knowledge and continuously adjust to changing conditions. Addressing these challenges is critical for applying forecasting models in real-world settings.

Building on these observations, recent work has begun exploring the paradigm of time series foundation models (TSFMs), inspired by their success in NLP and computer vision (Ruiz et al., 2024). These approaches aim to learn general-purpose representations that can transfer across tasks and domains. Nevertheless, building transferable foundation models for time series involves unique obstacles not present in NLP, including cross-domain heterogeneity and non-stationary temporal dynamics characterized by issues such as heteroskedasticity, unit roots, or intermittency (Ruiz et al., 2024). Most existing TSFMs rely on Transformer-based architectures, pre-trained with generative or contrastive objectives, with models such as Chronos (Ansari et al., 2024), Moirai (Woo et al., 2024), and TimeMOE (Shi et al., 2025) showing encouraging results. Nevertheless, important challenges remain: these models are computationally intensive, are often evaluated under narrow experimental conditions (e.g., homogeneous datasets or a single forecasting task), and their reliance on attention mechanisms raises questions about temporal inductive biases (Zeng et al., 2022).

To address these limitations, we introduce `TimeGEN` (Temporal Generative Encoder Network), a novel generative deep learning architecture specifically designed for TL in time series forecasting. An overview of the architecture is shown in Figure 1. Unlike efforts toward general-purpose foundation models, our aim is more focused: cross-domain transferability within a design that is both efficient and lightweight. We concentrate on the forecasting task, where TL is both highly practical and

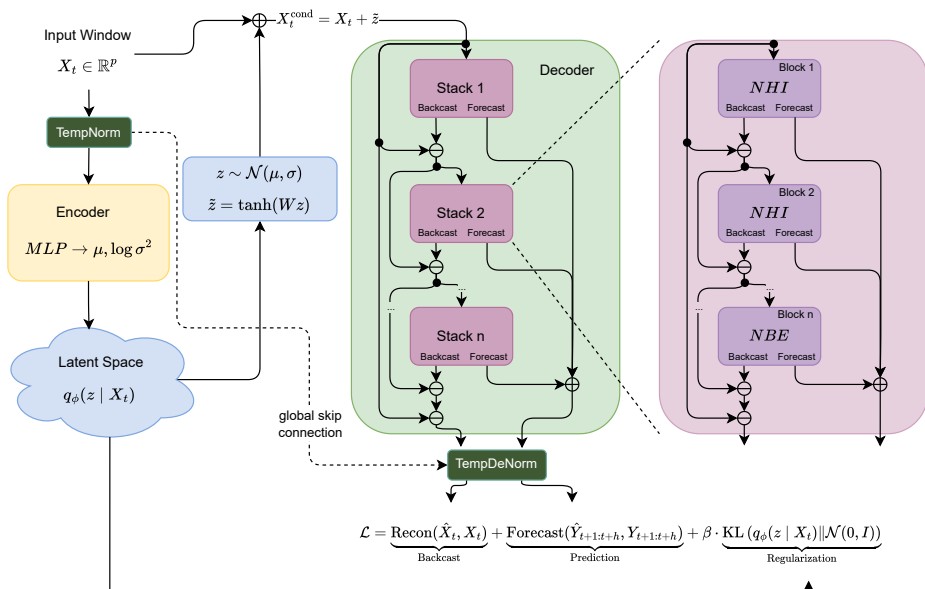

Figure 1: `TimeGEN` normalizes the input window $X_t$, then encodes it into a diagonal Gaussian posterior over latent vector $z$ via an MLP. A bounded projection $\tilde{z}$ is concatenated with $X_t$ and passed to each decoder block. The decoder is a residual stack of lightweight MLP blocks (NHI or NBE) that produce partial backcasts and forecasts. These outputs are then denormalized using the same instance-wise mean and variance from the initial normalization, passed via a global skip connection. Forecasts are summed up, and the input is reconstructed from the final residual. Training minimizes a combined loss over reconstruction, forecasting, and Kullback–Leibler (KL) divergence.

underexplored. In doing so, we deliberately explore alternatives to Transformer-based designs, whose high computational cost and limited temporal inductive bias pose challenges for scalable time series applications (Zeng et al., 2022).

At its core, a variational encoder learns high-level temporal representations that generalize across diverse series and domains. These representations are shaped by a combined reconstruction and forecasting objective, which aligns the latent space to preserve local detail while capturing global predictive dependencies. To further ensure stability and robustness under varying scales and noise conditions, we integrate temporal normalization layers (Kim et al., 2022). To capture dynamics at multiple resolutions, we employ a modular decoder that integrates neural basis expansion (Oreshkin et al., 2020) for sharp, high-frequency patterns with multi-rate interpolation blocks (Challu et al., 2023) for smooth, long-range trends. Conditioned on the global latent representation, this design captures shared structure across domains and adapts to task-specific variations, while remaining lightweight, fully MLP-based, and computationally efficient.

We evaluate `TimeGEN` on ten heterogeneous datasets comprising over 60 million observations, covering a broad range of domains, sampling frequencies, and temporal dynamics. Experiments span four evaluation regimes: (i) full-shot, (ii) in-domain zero-shot, (iii) single-source out-of-domain zero-shot, and (iv) multi-source out-of-domain zero-shot. The latter three regimes are designed to evaluate generalization under distributional shift, a core challenge in time series forecasting.

`TimeGEN` consistently outperforms SOTA models, including NHITS (Challu et al., 2023), KAN (Liu et al., 2025), PatchTST (Nie et al., 2023), iTransformer (Liu et al., 2023), TSMixer (Chen et al., 2023), TFT (Lim et al., 2021), and TimeMOE (Shi et al., 2025). It also delivers substantially better computational efficiency than all SOTA methods.

Our main contributions are:

1. A generative approach designed for TL in time series forecasting. `TimeGEN` employs a variational encoder to learn high-level temporal representations that generalize across series and domains, enabling strong zero-shot and cross-domain performance. A joint reconstruction–forecasting loss and temporal normalization further promote stable training and robust transfer.

2. A lightweight multiscale decoder architecture. A modular residual stack combines components to capture both short-term variations and long-range trends. Conditioned on a global latent vector, the decoder remains fully MLP-based, efficient, and well-suited for diverse forecasting domains.

3. Extensive empirical validation across domains. We evaluate `TimeGEN` on ten publicly available datasets, covering both conventional and transfer learning regimes. `TimeGEN` consistently outperforms SOTA methods in zero-shot and cross-domain settings, reducing forecasting error by between 8.2% and 38.2% compared to leading SOTA approaches and achieves a 2–30$\times$ speedup in training time.

To facilitate reproducibility and future research, we make `TimeGEN` publicly available along with all code, configurations, and datasets used in our experiments.[1]

## 2 BACKGROUND

**Time Series Forecasting.** Let $\mathcal{Y} = \{Y^{(i)}\}_{i=1}^n$ denote a collection of $n$ univariate time series, where each series $Y^{(i)} = \{y_t^{(i)}\}_{t=1}^{T_i}$ consists of scalar observations indexed by time $t$, and $T_i$ is the length of the $i$-th series. Such datasets are common in applications like retail forecasting, traffic prediction, and sensor monitoring, where each sequence may vary in length but shares structural or statistical similarities with others.

The goal of global forecasting is to learn a single predictive model across all time series in $\mathcal{Y}$, leveraging shared patterns to improve accuracy and data efficiency (Januschowski et al., 2020; Godahewa et al., 2021). Given a context window of length $p$, the task is to predict the next $h$ future values. For each time series $Y^{(i)}$, we construct training examples using time-delay embedding (Bontempi et al., 2013), yielding input-output pairs of the form:

$$X_t^{(i)} = \{y_{t-p+1}^{(i)}, \ldots, y_t^{(i)}\} \in \mathbb{R}^p, \quad Y_{t+1:t+h}^{(i)} = \{y_{t+1}^{(i)}, \ldots, y_{t+h}^{(i)}\} \in \mathbb{R}^h.$$

The model is trained to approximate a global forecasting function:

$$f_\theta : \mathbb{R}^p \to \mathbb{R}^h, \quad \hat{Y}_{t+1:t+h}^{(i)} = f_\theta(X_t^{(i)}).$$

To build the training set, we extract sliding windows from all series in the collection:[2]

$$\mathcal{D}_{\text{train}} = \bigcup_{i=1}^n \left\{ (X_t^{(i)}, Y_{t+1:t+h}^{(i)}) \mid p \le t \le T_i - h \right\}.$$

We distinguish three common ways to partition time series data for training and evaluation, always ensuring that training and testing use comparable partitions, regardless of the source. In the first case, $\mathcal{Y}_{\text{train}}, \mathcal{Y}_{\text{test}} \subset Y^{(i)}$, the model is trained and tested on different temporal segments of the same series. In the second, $\mathcal{Y}_{\text{train}}, \mathcal{Y}_{\text{test}} \subset \mathcal{D}^{(j)}$, with disjoint sets of series, representing training and testing on different time series from the same dataset. In the third, $\mathcal{Y}_{\text{train}} \subset \mathcal{D}^{(j)}$, $\mathcal{Y}_{\text{test}} \subset \mathcal{D}^{(k)}$, with $j \ne k$, training and testing are performed on series from different datasets. We treat each dataset $\mathcal{D}^{(j)} \subset \mathcal{Y}$ as representing a distinct domain $\mathcal{S}_j$, characterized by shared temporal or contextual properties (e.g., frequency, application, or structure).

---

[1]Implementation and experiments available at `https://anonymous.4open.science/r/TimeGEN-0296`

[2]When unambiguous, we omit the superscript $(i)$ for clarity.

**Deep Learning for Time Series Forecasting.** Deep learning has significantly advanced time series forecasting by enabling models to capture complex temporal patterns. Many approaches follow an autoregressive framework, predicting future values from fixed-length past contexts. Recurrent neural networks (RNNs), such as Long Short-Term Memory (LSTM) networks and Gated Recurrent Units (GRUs), have been widely adopted for their ability to model sequential dependencies (Yamak et al., 2019), with enhancements such as dilated connections (Chang et al., 2017) and probabilistic forecasting in models such as DeepAR (Salinas et al., 2020). Transformer-based architectures, including TFT (Lim et al., 2021), iTransformer (Liu et al., 2023), and PatchTST (Nie et al., 2023), apply attention mechanisms to capture long-range dependencies without recurrence. Despite strong empirical results in some settings, concerns have been raised about their reliability (Zeng et al., 2023), including temporal information loss and sensitivity to hyperparameters.

Simpler architectures (e.g. MLP-based models) have recently demonstrated competitive performance. N-BEATS (Oreshkin et al., 2019) and NHITS (Challu et al., 2023) use residual and hierarchical designs to model trends and seasonality, while TSMixer (Chen et al., 2023) introduces time-feature mixing for efficient long-sequence modeling. Convolutional approaches, such as Temporal Convolutional Networks (TCNs) (Bai et al., 2018), use dilated convolutions for parallel, long-range processing. More recently, Kolmogorov–Arnold Networks (KANs) (Liu et al., 2025) have emerged as alternatives to MLPs, using learnable univariate transformations inspired by the Kolmogorov–Arnold representation theorem.

**Foundation Models for Time Series.** Foundation models have recently extended to time series, aiming to generalize across tasks and domains with minimal fine-tuning. Inspired by successes in NLP and computer vision, these time series foundation models (TSFMs) learn general-purpose temporal representations through large-scale pre-training. A recent taxonomy by Ruiz et al. (2024) classifies TSFMs by modality, architecture, pre-training, and adaptation strategy.

Transformer-based TSFMs typically use self-supervised generative pre-training, with models such as PatchTST (Nie et al., 2023), Moirai (Woo et al., 2024), Chronos (Ansari et al., 2024), and TimeMOE (Shi et al., 2025) using attention and token-mixing to model long-range dependencies. MLP-based TSFMs such as TSMixer (Chen et al., 2023) adopt simpler architectures while still achieving strong performance, typically following a generative pre-training approach. Fully supervised strategies also exist, such as TTMs (Ekambaram et al., 2024), a compact TL model based on TSMixer. Contrastive pre-training has also been explored in models such as TS2Vec (Yue et al., 2022), which learns time series representations by maximizing similarity between temporally coherent segments.

## 3 TimeGEN

We introduce `TimeGEN` (Temporal Generative Encoder Network), a generative, lightweight, and fully MLP-based architecture for time series forecasting.

The architecture has three key components: a variational encoder, a lightweight decoder stack, and a training objective that jointly optimizes reconstruction, forecasting, and regularization. To improve robustness under distribution shifts, we use temporal normalization. Figure 1 provides an overview of the architecture.

### 3.1 Latent Temporal Encoder

Given an input window of length $p$, denoted $X_t = \{y_{t-p+1}, \ldots, y_t\}$, we represent this window as a fixed-length vector $X_t \in \mathbb{R}^p$. The encoder maps this vector to the parameters of a multivariate Gaussian distribution over the latent variable $z$:

$$q_\phi(z \mid X_t) = \mathcal{N}\big(\mu, \mathrm{diag}(\sigma^2)\big),$$

where

$$(\mu, \log \sigma^2) = g_\phi(X_t).$$

Here, $g_\phi$ denotes the encoder function, implemented as an MLP. It outputs a mean vector $\mu \in \mathbb{R}^d$ and a log-variance vector $\log \sigma^2 \in \mathbb{R}^d$. Using the reparameterization trick (Kingma & Welling, 2022), we sample:

$$z = \mu + \sigma \odot \epsilon, \quad \epsilon \sim \mathcal{N}(0, I).$$

The latent variable $z$ summarizes the temporal structure and variability observed in the input window. We interpret it as a global context that captures cross-series and cross-domain information useful for forecasting. This latent representation is transformed via a linear projection followed by a nonlinearity via $\tilde{z} = \tanh(Wz)$ to produce a bounded latent embedding $\tilde{z} \in \mathbb{R}^p$.

The latent embedding $\tilde{z}$ is concatenated with the input window to form the latent-conditioned input $X_t^{\text{cond}} = [X_t, \tilde{z}]$, where $[X_t, \tilde{z}]$ denotes feature-wise concatenation. This conditioning signal is passed to each decoder block, providing shared global context.

## 3.2 LATENT-CONDITIONED DECODER

The decoder of `TimeGEN` is a residual stack composed of $L$ blocks, each producing a partial forecast and a backcast. The architecture follows the design of recent residual forecasting models (Oreshkin et al., 2020; Challu et al., 2023), but unlike those models, each block in `TimeGEN` is conditioned on the global latent vector $z$, which provides shared context across the stack.

Formally, the $\ell$-th block is denoted by $h^{(\ell)}$, and takes as input the residual signal $r^{(\ell)}$. It outputs a backcast $b^{(\ell)}$ and a forecast $f^{(\ell)}$:

$$(b^{(\ell)}, f^{(\ell)}) = h^{(\ell)}(r^{(\ell)}, z),$$
$$r^{(\ell+1)} = r^{(\ell)} - b^{(\ell)}.$$

The input for the first block is the latent-conditioned input $X^{\text{cond}}$, reversed in time. Forecasts from all blocks are accumulated to form the final prediction:

$$\hat{Y}_{t+1:t+h} = \sum_{\ell=1}^{L} f^{(\ell)}.$$

The final residual $r^{(L)}$ represents the portion of the input that remains unexplained after all backcasts have been subtracted. Since each block removes a partial backcast from the residual, the sum of all backcasts is implicitly equal to the difference between the original input and the final residual. Thus, the input is reconstructed with a single subtraction:

$$\hat{X}_t = X_t^{\text{cond}} - r^{(L)}.$$

To capture a broad range of temporal patterns, the decoder interleaves two types of blocks (Figure 1 illustrates this):

- **Neural hierarchical interpolation (NHI) blocks** (Challu et al., 2023) operate on down-sampled views of the residual and learn to interpolate back to the original resolution. This enables them to efficiently model smooth, low-frequency components such as long-term trends or seasonal effects, capturing global structure with fewer parameters.

- **Neural basis expansion (NBE) blocks** (Oreshkin et al., 2020) use flexible, learned basis expansions to reconstruct sharper, high-frequency variations. These include irregular patterns, short-term fluctuations, and noise components that are difficult to capture through interpolation alone.

This hierarchical structure allows the decoder to represent both global regularities and local details in a modular way. The two block types can be interleaved, with both the number of blocks of each type and their ordering treated as tunable parameters. Moreover, these blocks are MLP-based and thus less computationally expensive than attention or recurrence mechanisms.

### 3.3 TRAINING OBJECTIVE

`TimeGEN` is trained end-to-end with a joint objective that combines reconstruction, forecasting, and variational regularization. Let $\hat{X}_t$ denote the reconstruction of the input window, and $\hat{Y}_{t+1:t+h}$ the predicted future values. The total loss is defined as:

$$\mathcal{L} = \underbrace{\text{Recon}(\hat{X}_t, X_t)}_{\text{Backcast}} + \underbrace{\text{Forecast}(\hat{Y}_{t+1:t+h}, Y_{t+1:t+h})}_{\text{Prediction}} + \beta \cdot \underbrace{\text{KL}\left(q_\phi(z \mid X_t) \parallel \mathcal{N}(0, I)\right)}_{\text{Regularization}}.$$

The reconstruction loss is computed using the final residual $r^{(L)}$, which captures the remaining unexplained signal after all backcast components have been removed. By penalizing this residual, the model is forced to minimize any leftover error, requiring the decoder stack and the latent representation $z$ to jointly account for the full input sequence. This residual-based formulation also acts as an implicit regularizer. Because each stage must only explain what remains, the model is naturally encouraged to build structured latent representations and to produce a clean, multi-scale decomposition in the decoder.

The forecasting loss complements this by explicitly training the model to anticipate future dynamics. Optimizing over future horizons prevents the network from overfitting to short-term reconstructions and encourages latent variables to capture predictive temporal structure, which is essential for robust generalization in time-series modeling.

We use the mean absolute error for both the reconstruction and forecasting losses. The KL divergence term regularizes the latent space by encouraging the approximate posterior $q_\phi(z \mid X)$ to remain close to a standard Gaussian prior, which improves generalization and stabilizes the latent representations. The KL weight $\beta$ is linearly annealed during training to prevent latent collapse (Burgess et al., 2018).

## 4 EXPERIMENTAL SETUP

We evaluate `TimeGEN` under conventional and TL conditions, comparing its performance to SOTA approaches, including recent Transformer and MLP-based architectures, on ten datasets. We focus on five research questions:

- **Q1**: How does `TimeGEN` perform in standard full-shot forecasting settings?
- **Q2**: How well does it generalize to unseen time series from the same domain?
- **Q3**: Can it effectively transfer to new domains when trained on a single source dataset?
- **Q4**: Can it effectively transfer to new domains when trained on multiple datasets and domains?
- **Q5**: Does its architecture offer advantages in training efficiency?
- **Q6**: Which architectural components are most critical for the performance of `TimeGEN`?

### 4.1 DATASETS

We use a diverse set of ten benchmark datasets covering a wide range of domains, sampling frequencies (monthly, quarterly, yearly, and daily), and forecasting horizons. These datasets include demand forecasting, economic indicators, tourism trends, and traffic patterns, among others. They reflect realistic and heterogeneous time series applications. Specifically, we use the Tourism (Athanasopoulos et al., 2011), M1 (Makridakis et al., 1982) (monthly and quarterly), M3 (Makridakis & Hibon, 2000) (monthly, quarterly and yearly), M4 (Makridakis et al., 2020) (monthly and quarterly), M5 (Makridakis et al., 2022), and Traffic (Olivares et al., 2024) datasets. In total, the benchmark suite comprises over 106,000 time series and more than 60 million observations. This scale enables robust evaluation across a variety of conditions. A full summary of dataset characteristics is provided in Appendix A.

### 4.2 METHODS

We compare `TimeGEN` to seven SOTA methods with diverse architectural designs:

- NHITS (Challu et al., 2023): A hierarchical MLP-based model combining residual forecasting with multi-rate input sampling, achieving strong performance on long-horizon tasks.

- KAN (Liu et al., 2025): An approach based on Kolmogorov–Arnold Networks that replaces fixed activations with learnable univariate transformations for greater expressiveness.

- PatchTST (Nie et al., 2023): A transformer-based model that segments input sequences into non-overlapping patches and applies self-attention over these patches for efficient long-sequence forecasting.

- iTransformer (Liu et al., 2023): A transformer variant for time series that leverages instance normalization and token mixing to enhance stability and performance on long-context forecasting tasks.

- TSMixer (Chen et al., 2023): A lightweight MLP architecture alternating mixing operations over time and feature dimensions. It is designed to be efficient while maintaining high accuracy across diverse tasks.

- TFT (Lim et al., 2021): The Temporal Fusion Transformer integrates attention, gating mechanisms, and variable selection, providing interpretable multi-step forecasts.

- TimeMOE (Shi et al., 2025): A mixture-of-experts architecture for time series forecasting, where specialized experts are dynamically routed to different inputs through a gating network.

We use the *neuralforecast* library from Nixtla to implement and optimize all methods. For architectures not originally available in *neuralforecast*, such as TimeMOE, we implemented and integrated them by adapting the official code released by the authors.

For all algorithms, we perform hyperparameter optimization using random search. From a predefined pool of possible configurations, we randomly sample and evaluate 20 configurations for each method using a validation set. The optimization process includes both model-specific parameters and basic data preprocessing choices (e.g., normalization strategy). The best-performing configuration is then used to retrain the model on the complete training data.

We rely on the default hyperparameter search spaces defined in the implementation of each method and adopt analogous definitions for TimeGEN wherever possible. For TimeMOE, we define the search space of hyperparameters based on the available configuration for the 50M parameter pretrained version released on Hugging Face.[3] All experiments were conducted using a setup with a Quadro RTX 8000 GPU (48 GB VRAM), an Intel Xeon W-2295 CPU (3.0 GHz), and 128 GB RAM. For complete details, refer to Appendix F.

### 4.3 EVALUATION SETTINGS

We evaluate TimeGEN and other SOTA methods under the four evaluation settings illustrated in Figure 2. Besides the standard task-specific setting, we also evaluate the models in three zero-shot TL scenarios. These settings test the ability of the models to generalize to unseen series or domains:

- **Full-shot** refers to the standard task-specific setup where training and testing are performed on different time segments of the same time series. Formally, $\mathcal{Y}_{\text{train}}, \mathcal{Y}_{\text{test}} \subset Y^{(i)}$ for each $Y^{(i)} \in \mathcal{D}^{(j)}$.

- **In-domain transfer learning** uses disjoint subsets of time series from the same dataset for training and testing, i.e., $\mathcal{Y}_{\text{train}}, \mathcal{Y}_{\text{test}} \subset \mathcal{D}^{(j)}$ with $\mathcal{Y}_{\text{train}} \cap \mathcal{Y}_{\text{test}} = \emptyset$. This evaluates generalization to unseen series within a single domain.

- **Single source out-of-domain transfer learning** evaluates cross-domain generalization when the model is trained on one dataset and tested on another. That is, $\mathcal{Y}_{\text{train}} \subset \mathcal{D}^{(j)}$, $\mathcal{Y}_{\text{test}} \subset \mathcal{D}^{(k)}$, with $j \neq k$.

- **Multiple source out-of-domain transfer learning** simulates the foundation model setting, where training is performed on multiple datasets and testing on a previously unseen domain.

---

[3]https://huggingface.co/Maple728/TimeMoE-50M

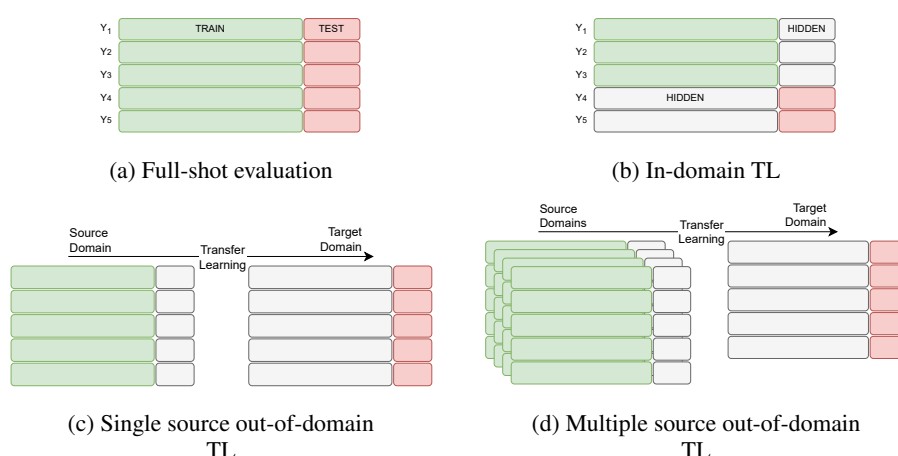

Figure 2: Evaluation settings for time series forecasting experiments: (a) traditional train-test split within time series, (b) generalization to unseen series within the same domain, (c) cross-domain transfer from a single source, and (d) multi-source transfer to an unseen domain. Color coding indicates data usage: green for training, red for testing, and grey for unused data.

Table 1: MASE and average rank (lower is better) across datasets and evaluation settings. Best and second-best values are **bolded** and underlined. The *Time* column reports normalized training time relative to the fastest method (TimeGEN = 1.0).

| Method | Time (× TimeGEN) | Full-shot | | In-domain | | Single-source | | Multi-source | |
|---|---|---|---|---|---|---|---|---|---|
| | | MASE | Rank | MASE | Rank | MASE | Rank | MASE | Rank |
| **TimeGEN** | 1.0 | 1.332 | 2.7 | **1.409** | 3.8 | **1.926** | **3.0** | **1.493** | **1.8** |
| KAN | 2.076 | 1.376 | 3.3 | 1.442 | 3.2 | 10.962 | 4.7 | 1.627 | 3.5 |
| NHITS | 3.507 | **1.319** | **2.3** | 1.423 | **2.8** | 161.73 | 5.0 | 1.783 | 4.6 |
| PatchTST | 5.235 | 1.432 | 4.6 | 1.438 | 4.4 | 1.986 | 3.5 | 1.681 | 4.2 |
| TFT | 4.105 | 1.439 | 4.1 | 1.474 | 4.3 | 2.163 | 3.9 | 1.714 | 4.2 |
| TSMixer | 31.042 | 1.810 | 7.2 | 1.676 | 6.5 | 2.160 | 5.0 | 1.816 | 6.2 |
| TimeMOE | 28.082 | 1.445 | 4.7 | 1.443 | 4.2 | 2.255 | 5.1 | 1.837 | 5.4 |
| iTransformer | 26.760 | 1.667 | 7.0 | 1.632 | 6.4 | 2.429 | 5.4 | 2.416 | 5.1 |

Let $\{\mathcal{D}^{(1)}, \ldots, \mathcal{D}^{(m)}\}$ be the full set of datasets. For each held-out dataset $\mathcal{D}^{(k)}$, we define $\mathcal{Y}_{\text{train}} = \bigcup_{j \neq k} \mathcal{D}^{(j)}$ and $\mathcal{Y}_{\text{test}} = \mathcal{D}^{(k)}$.

We evaluate forecasting performance using the Mean Absolute Scaled Error (MASE) (Hyndman & Athanasopoulos, 2021), a scale-independent metric that enables consistent comparison across time series of varying magnitudes and units. Unlike percentage-based metrics such as SMAPE, MASE is robust when target values are near zero, avoiding division-by-zero issues and inflated error scores (Makridakis et al., 2020). This makes it particularly suitable for heterogeneous forecasting benchmarks. The full formula and details are provided in Appendix G.

## 5 RESULTS AND DISCUSSION

We evaluate all methods under the four regimes introduced in Section 4.3. Table 1 summarizes MASE and average rank across datasets, while Appendix B–E reports results per dataset.

In the full-shot setting (**Q1**), TimeGEN performs on par with the strongest baselines. NHITS remains the top method in terms of raw MASE, but TimeGEN achieves the second-best overall performance and rank. This is an important result, given that we designed TimeGEN specifically for TL rather than purely within-domain optimization. By contrast, models like TSMixer and Transformer-based approaches (TimeMOE, iTransformer) perform worse, highlighting the limitations of heavier architectures when trained from scratch.

Table 2: Ablation study of `TimeGEN`. Each variant removes or modifies one key component. Results are reported in terms of MASE (lower is better).

| Variant | Full-shot | In-domain | Single-source |
|---|---|---|---|
| **TimeGEN** | 1.33 | 1.41 | **1.93** |
| Shallow Latent | **1.31** | **1.32** | 2.50 |
| Shallow + No Mixture | 1.33 | 1.33 | 2.72 |
| No Normalization | 1.32 | 1.33 | 5.97 |

In the in-domain zero-shot regime (**Q2**), `TimeGEN`, KAN, and NHITS deliver competitive performances. Interestingly, `TimeGEN` obtains the best MASE while KAN and especially NHITS obtain lower average ranks. Some models, such as PatchTST and TimeMOE, matched their full-shot performance, and TSMixer even surpassed it, which was unexpected. Despite these cases, `TimeGEN`, KAN, and NHITS consistently outperformed the rest.

The single-source cross-domain regime (**Q3**) is the most challenging: models trained on one dataset are directly evaluated in another. In this setup, `TimeGEN` clearly leads with the lowest MASE and the best average rank, while other methods degrade substantially. In this setting, NHITS shows a sharp increase in error, and KAN also degrades noticeably. PatchTST was the only model with results close to `TimeGEN` in both MASE and rank. These results suggest that `TimeGEN` is well-suited for TL across domains, even in settings with very limited training data.

Finally, in the multi-source regime (**Q4**), where training spans multiple domains before testing on a held-out one, `TimeGEN` achieves the best performance again in terms of both MASE and rank. It outperforms Transformer-based models like PatchTST and TimeMOE and shows that a lightweight architecture can generalize better across time series domains, even as the amount of training data increases. The relative improvement over competing SOTA methods ranges from 8.2% to 38.2%.

Beyond accuracy, efficiency is another critical aspect to assess (**Q5**). Normalized training times show that `TimeGEN` is the fastest method. It is more than twice as fast as KAN, three times faster than NHITS, and an order of magnitude faster than most Transformer-based models. TSMixer, TimeMOE, and iTransformer require 25–30$\times$ longer training times. This makes `TimeGEN` especially practical for scenarios where frequent retraining or deployment across many domains is required.

**Ablation study.**    To assess the contribution of individual components (**Q6**), Table 2 reports an ablation study. Removing deep latent conditioning slightly improves full-shot performance and yields the strongest in-domain generalization, but leads to clear losses in cross-domain transfer. Building on this, removing both deep latent conditioning and the mixture of blocks in the decoder increases error across all regimes, which shows the importance of the flexibility that this component provides. Finally, keeping latent conditioning and mixture but omitting temporal normalization produces outliers in TL, with large errors in some cases, showing its role in stabilizing domain adaptation.

## 6 CONCLUSION

This work introduces `TimeGEN`, a generative model for time series forecasting that combines a variational encoder with a lightweight, MLP-based decoder. Unlike most existing approaches that rely on attention mechanisms or extensive pre-training, `TimeGEN` achieves strong generalization across domains through a compact, efficient design trained entirely from scratch.

Through extensive experiments on ten diverse datasets and four evaluation regimes, we showed that `TimeGEN` outperforms SOTA methods in zero-shot and cross-domain scenarios while remaining highly competitive in full-shot settings. It achieves these results with significantly lower training time, up to 30$\times$ faster than leading transformer and MLP-based methods. It demonstrates its practical advantages in both research and real-world applications.

We believe `TimeGEN` highlights the potential for developing simpler and more efficient foundation models that generalize across domains without relying heavily on model complexity or large-scale pre-training.

**Reproducibility Statement.** We aim to make all experiments fully reproducible. Our implementation, training/evaluation scripts, and configuration files are provided in the anonymous repository linked in the main text (see footnote in Section 1). Dataset composition, splits, horizons, and frequencies are described in Section 4 and summarized in Appendix A. Preprocessing (including normalization choices) and windowing are specified there as well. Hyperparameter search spaces, optimization details, and model settings for `TimeGEN` and all SOTA methods are documented in Section 4.2 and Appendix F. We use a fixed random seed per trial and report the best validation run selected from 20 random configurations, then retrain with the selected configuration. Evaluation protocols and metrics (MASE), including formulas and seasonal-period choices, are provided in Section 4.3 and Appendix G. Hardware and software environments (e.g., GPU/CPU, memory) are reported in Section 4.3. The repository contains instructions for rerunning all experiments for the different evaluation regimes (Figure 2).

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

## A    DATASET SUMMARY

Table 3 provides a detailed summary of the ten benchmark datasets used in our experiments. The datasets span a range of domains and include varying temporal frequencies (monthly, quarterly, yearly, and daily) and forecasting horizons. Some datasets, such as M1, M3, and M4, include multiple frequency variants. This diversity ensures that the benchmark suite reflects a broad spectrum of real-world forecasting scenarios, including differences in scale, seasonality, sparsity, and volatility.

## B    FULL-SHOT RESULTS

Table 4 presents MASE for the full-shot setting (training and testing on the same dataset).

## C    IN-DOMAIN RESULTS

Table 5 presents MASE for the in-domain generalization setting, where models are evaluated on unseen time series from the same dataset.

## D    SINGLE-SOURCE OUT-OF-DOMAIN RESULTS

Table 6 reports the MASE for the single-source out-of-domain setting. In this setup, each model is trained on a single source dataset and evaluated on all remaining datasets to assess cross-domain generalization. We report the average MASE per source dataset by aggregating performance across all its corresponding target evaluations. This provides a robust estimate of how well each model transfers when trained on a specific domain.

## E    MULTI-SOURCE OUT-OF-DOMAIN RESULTS

In the multi-source out-of-domain setting, a model is trained on the union of all datasets *except one*, which is held out entirely for evaluation. This setup simulates a real-world deployment scenario,

Table 3: Summary of the datasets used in the experimental setup, including the number of time series, number of observations, forecast horizon, and frequency. Sources: Tourism (Athanasopoulos et al., 2011), M1 (Makridakis et al., 1982), M3 (Makridakis & Hibon, 2000), M4 (Makridakis et al., 2020), M5 (Makridakis et al., 2022), Traffic (Olivares et al., 2024).

| Dataset | Frequency | # time series | # observations | Horizon ($H$) |
|---|---|---|---|---|
| Tourism | Monthly | 366 | 109,661 | 24 |
| M1 | Monthly | 617 | 44,884 | 24 |
| | Quarterly | 203 | 8,323 | 8 |
| M3 | Monthly | 1,428 | 167,475 | 24 |
| | Quarterly | 756 | 37,026 | 8 |
| | Yearly | 645 | 18,333 | 4 |
| M4 | Monthly | 48,000 | 11,246,400 | 24 |
| | Quarterly | 23,999 | 2,398,978 | 8 |
| Traffic | Daily | 207 | 75,762 | 30 |
| M5 | Daily | 30,490 | 47,624,269 | 30 |
| **Total** | | **106,711** | **61,731,111** | – |

Table 4: Full-shot results (MASE). Best and second-best values are **bolded** and underlined. Lower is better.

| Method | M1 | | M3 | | | M4 | | M5 | Tourism | Traffic |
|---|---|---|---|---|---|---|---|---|---|---|
| | Monthly | Quarterly | Monthly | Quarterly | Yearly | Monthly | Quarterly | Daily | Monthly | Daily |
| **TimeGEN** | 1.578 | 1.762 | 1.088 | **1.096** | **2.256** | **1.119** | 1.249 | 0.844 | **1.463** | 0.868 |
| KAN | 1.597 | 1.895 | 1.070 | 1.332 | 2.287 | 1.206 | 1.212 | **0.791** | 1.521 | 0.836 |
| NHITS | **1.523** | 1.802 | **1.031** | 1.144 | 2.347 | 1.122 | **1.166** | 0.796 | 1.478 | **0.764** |
| PatchTST | 1.759 | 1.785 | 1.204 | 1.232 | 2.266 | 1.212 | 1.325 | 0.810 | 1.684 | 1.031 |
| TFT | 1.734 | **1.603** | 1.128 | 1.332 | 2.482 | 1.252 | 1.290 | 0.797 | 1.942 | 0.812 |
| TSMixer | 3.503 | 1.947 | 1.228 | 1.678 | 2.814 | 1.322 | 1.760 | 0.831 | 2.191 | 0.814 |
| TimeMOE | 1.918 | 1.671 | 1.166 | 1.443 | 2.325 | 1.263 | 1.413 | 0.803 | 1.526 | 0.921 |
| iTransformer | 2.367 | 1.913 | 1.375 | 1.682 | 2.520 | 1.664 | 1.535 | 0.829 | 1.863 | 0.911 |

where a forecasting model must generalize to a previously unseen domain based on diverse prior training data. It also mirrors the pre-training paradigm commonly used with large Transformer-based models.

Table 7 reports the MASE on each held-out dataset. For example, when `M1 - Monthly` is used as the test set, the model is trained on all other datasets.

## F   IMPLEMENTATION AND HYPERPARAMETER DETAILS

All non-pre-trained models are instantiated using the default hyperparameter configurations provided by the `neuralforecast` library via their `auto` wrappers and trained through a custom `ModelPipeline`. For each model, we perform a *random search* (20 trials, `BasicVariantGenerator(seed=1)`) on the training set. The trial minimizing validation MASE is selected, and the corresponding model is then retrained on the combined train and validation data before final evaluation.

**Shared hyperparameter search space:**

- `learning_rate` $\sim \log U[10^{-5}, 10^{-2}]$
- `batch_size` $\in \{32, 64, 128, 256\}$
- `windows_batch_size` $\in \{128, 256, 512, 1024\}$
- `random_seed` $\in [1, 20]$

**TimeGEN**

Table 5: In-domain forecasting results (MASE). Best and second-best values are **bolded** and underlined. Lower is better.

| Method | M1 | | M3 | | | M4 | | M5 | Tourism | Traffic |
|---|---|---|---|---|---|---|---|---|---|---|
| | Monthly | Quarterly | Monthly | Quarterly | Yearly | Monthly | Quarterly | Daily | Monthly | Daily |
| **TimeGEN** | 1.800 | 2.316 | 1.065 | **1.113** | 2.294 | **1.086** | 1.267 | 0.840 | 1.445 | 0.864 |
| KAN | **1.676** | 2.708 | 1.017 | 1.161 | **2.206** | 1.223 | 1.224 | **0.785** | 1.495 | 0.927 |
| NHITS | 2.531 | 2.009 | **0.965** | 1.125 | 2.256 | 1.134 | **1.150** | 0.792 | 1.504 | **0.761** |
| PatchTST | 1.788 | 2.021 | 1.120 | 1.316 | 2.308 | 1.162 | 1.304 | 0.790 | 1.763 | 0.807 |
| TFT | 1.804 | **1.907** | 1.092 | 1.270 | 2.265 | 1.176 | 1.296 | 0.790 | 1.982 | 1.161 |
| TSMixer | 1.925 | 2.258 | 1.235 | 1.420 | 3.079 | 1.292 | 1.592 | 0.818 | 2.338 | 0.799 |
| TimeMOE | 1.982 | 1.921 | 1.026 | 1.298 | 2.388 | 1.271 | 1.292 | 0.810 | **1.421** | 0.876 |
| iTransformer | 1.989 | 2.214 | 1.187 | 1.660 | 2.382 | 1.626 | 2.200 | 0.823 | 1.817 | 0.772 |

Table 6: Single-source out-of-domain forecasting results (MASE). Best and second-best values are **bolded** and underlined. Lower is better.

| Method | M1 | | M3 | | | M4 | | M5 | Tourism | Traffic |
|---|---|---|---|---|---|---|---|---|---|---|
| | Monthly | Quarterly | Monthly | Quarterly | Yearly | Monthly | Quarterly | Daily | Monthly | Daily |
| **TimeGEN** | 1.857 | **1.814** | 1.516 | 1.932 | 1.536 | 1.646 | **1.728** | 3.155 | 1.797 | 2.166 |
| KAN | **1.642** | 2.142 | 1.593 | 16.142 | 45.966 | 20.025 | 20.236 | 3.228 | **1.789** | 4.220 |
| NHITS | 2.218 | 2.019 | 4.341 | 2.541 | 1854.614 | 2.038 | 2.396 | 3.451 | 4.610 | 2.221 |
| PatchTST | 1.703 | 2.329 | 1.567 | 1.867 | **1.530** | 1.714 | 1.809 | 3.411 | 1.810 | 2.013 |
| TFT | 1.665 | 2.097 | 1.608 | 2.575 | 2.197 | 1.749 | 2.057 | 3.836 | 1.777 | 2.144 |
| TSMixer | 1.930 | 2.312 | 2.000 | **1.799** | 1.631 | 1.891 | 2.273 | 3.652 | 1.872 | 2.142 |
| TimeMOE | 1.997 | 3.057 | **1.368** | 2.269 | 5.063 | **1.513** | 2.253 | **1.242** | 3.918 | **1.902** |
| iTransformer | 2.001 | 2.159 | 1.694 | 2.089 | 1.845 | 3.037 | 3.224 | 4.388 | 1.706 | 2.066 |

- latent_dim $\in \{16, 32, 64, 128, 256\}$
- encoder_hidden_dims $\in \{[64, 32], [256, 128], [512, 256], [512, 256, 128]\}$
- n_beats_nblocks_stack_1, _2, _3 $\in \{0, 1\}$
- input_size $= H \times m$, $m \in \{1, 2, 3, 4, 5\}$
- n_pool_kernel_size $\in \{[2, 2, 1], 3 \times [1], 3 \times [2], 3 \times [4], [8, 4, 1], [16, 8, 1]\}$
- n_freq_downsample $\in \{[168, 24, 1], [24, 12, 1], [180, 60, 1], [60, 8, 1], [40, 20, 1], [1, 1, 1]\}$
- max_steps $\in [500, 1500]$, step $= 100$

**NHITS**

- input_size $= H \times m$, $m \in \{1, 2, 3, 4, 5\}$
- n_pool_kernel_size and n_freq_downsample as in TimeGEN
- scaler_type $\in \{\text{identity}, \text{standard}\}$
- max_steps $\in [500, 1500]$, step $= 100$

**KAN**

- input_size $= H \times m$, $m \in \{1, 2, 3, 4, 5\}$
- grid_size $\in \{5, 10, 15\}$
- spline_order $\in \{2, 3, 4\}$
- hidden_size $\in \{64, 128, 256, 512\}$
- scaler_type $\in \{\text{identity}, \text{standard}\}$
- max_steps $\in [500, 1500]$, step $= 100$

**PatchTST**

- input_size $= H \times m$, $m \in \{1, 2, 3\}$
- hidden_size $\in \{16, 128, 256\}$

Table 7: Multi-source out-of-domain forecasting results (MASE). Best and second-best values are **bolded** and underlined. Lower is better.

| Method | M1 | | M3 | | | M4 | | M5 | Tourism | Traffic |
|--------|---------|-----------|---------|-----------|-------|---------|-----------|-------|---------|--------|
| | Monthly | Quarterly | Monthly | Quarterly | Yearly | Monthly | Quarterly | Daily | Monthly | Daily |
| **TimeGEN** | **1.590** | **1.677** | **0.989** | **1.258** | 2.834 | **1.174** | **1.438** | **1.032** | **1.740** | 1.193 |
| KAN | 1.737 | 1.763 | 1.094 | 1.351 | 2.383 | 1.417 | 1.644 | 1.109 | 2.588 | 1.184 |
| NHITS | 1.862 | 1.953 | 1.090 | 1.396 | 2.721 | 1.522 | 1.512 | 2.310 | 2.368 | **1.092** |
| PatchTST | 1.696 | 2.004 | 1.129 | 1.454 | 2.559 | 1.260 | 1.539 | 1.094 | 2.307 | 1.132 |
| TFT | 1.750 | 1.763 | 1.205 | 1.433 | **2.340** | 1.297 | 1.476 | 1.134 | 3.130 | 1.460 |
| TSMixer | 1.885 | 2.111 | 1.247 | 1.564 | 3.001 | 1.276 | 1.641 | 1.303 | 2.529 | 1.602 |
| TimeMOE | 1.793 | 1.894 | 1.241 | 1.386 | 3.532 | 1.397 | 1.702 | 1.062 | 3.130 | 1.237 |
| iTransformer | 1.846 | 1.808 | 1.302 | 1.479 | 2.491 | 1.334 | 1.477 | 1.246 | 2.452 | 1.632 |

- n_heads $\in \{4, 16\}$

- patch_len $\in \{16, 24\}$

- revin $\in \{\text{False}, \text{True}\}$

- scaler_type $\in \{\text{identity}, \text{standard}\}$

- max_steps $\in \{500, 1000, 5000\}$

**iTransformer**

- input_size $= H \times m$, $m \in \{1, 2, 3, 4, 5\}$

- step_size $\in \{1, H\}$

- hidden_size $\in \{64, 128, 256\}$

- n_heads $\in \{4, 8\}$

- scaler_type $\in \{\text{identity}, \text{standard}\}$

- max_steps $\in \{500, 1000, 2000\}$

**TSMixer**

- input_size $= H \times m$, $m \in \{1, 2, 3, 4\}$

- step_size $\in \{1, H\}$

- n_block $\in \{1, 2, 4, 6, 8\}$

- ff_dim $\in \{32, 64, 128\}$

- dropout $\sim \mathcal{U}(0, 0.99)$

- scaler_type $\in \{\text{identity}, \text{standard}\}$

- max_steps $\in \{500, 1000, 2000\}$

**TFT**

- input_size $= H \times m$, $m \in \{1, 2, 3, 4, 5\}$

- step_size $\in \{1, H\}$

- hidden_size $\in \{64, 128, 256\}$

- n_head $\in \{4, 8\}$

- scaler_type $\in \{\text{identity}, \text{standard}\}$

- max_steps $\in \{500, 1000, 2000\}$

**TimeMOE**

- `hidden_size` $\in \{64, 128, 384\}$
- `intermediate_size` $\in \{128, 256, 512, 1536\}$
- `num_hidden_layers` $\in \{2, 4\}$
- `num_attention_heads` $\in \{2, 4, 8, 12\}$
- `num_experts` $\in \{2, 4, 8\}$
- `num_experts_per_tok` $\in \{1, 2\}$
- `attention_dropout` $\sim \mathcal{U}(0, 0.2)$
- `hidden_act` $\in \{$silu, gelu$\}$
- `num_key_value_heads` $\in \{1, 2, 4, 8\}$
- `rope_theta` $\in \{10000, 20000\}$
- `max_position_embeddings` $\in \{512, 4096\}$
- `router_aux_loss_factor` $\sim \log U[10^{-2}, 10^{-1}]$
- `input_size` $= H \times m, \ m \in \{1, 2, 3, 4, 5\}$
- `step_size` $\in \{1, H\}$
- `scaler_type` $\in \{$identity, standard$\}$
- `max_steps` $\in \{500, 1000, 2000\}$

## G    EVALUATION METRICS

We report forecasting accuracy using the Mean Absolute Scaled Error (MASE), defined as:

$$\text{MASE} = \frac{\frac{1}{h} \sum_{j=1}^{h} \left| \hat{y}_{t+j}^{(i)} - y_{t+j}^{(i)} \right|}{\frac{1}{T_i - m} \sum_{t'=m+1}^{T_i} \left| y_{t'}^{(i)} - y_{t'-m}^{(i)} \right|}$$

where $\hat{y}_{t+j}^{(i)}$ is the forecasted value and $y_{t+j}^{(i)}$ the true value at horizon step $j$, and $m$ is the seasonal period (e.g., 12 for monthly data with annual seasonality). The denominator normalizes the error by the average in-sample one-step naive forecast error, making MASE interpretable and comparable across datasets.

