# OpenReview forum: "TimeGEN: A Cross-Domain and Generative Model for Time Series Forecasting"
_ICLR.cc/2026/Conference — Submitted to ICLR 2026_

### Official Review · Reviewer_CsL5 · 2025-11-01

**Soundness:** 2
**Presentation:** 3
**Contribution:** 2
**Rating:** 2
**Confidence:** 4

**Summary:**

This paper proposes TimeGEN, a novel, lightweight, MLP-based architecture designed for transfer learning (TL) in time series forecasting. The primary goal is to create a model that excels in zero-shot and cross-domain scenarios, where it must make predictions on data from unseen domains.

**Strengths:**

1. The paper's main strength is its originality in creatively combining existing ideas (like VAEs and MLP-based models) into a novel, lightweight architecture. This hybrid model is specifically engineered to solve the difficult problem of cross-domain forecasting.

2. Another major strength is the quality of its experiments. The authors perform a rigorous evaluation across ten different datasets and, crucially, test the model in four distinct settings, including three challenging zero-shot and transfer-learning scenarios. This provides robust evidence for its claims.

3. The work is seen as highly significant for practical reasons. It successfully tackles poor generalization, a major bottleneck in forecasting. By doing this with a model that is 2-30x faster than competitors, it provides a compelling and efficient alternative to the "heavier is better" trend dominated by large Transformer models.

**Weaknesses:**

1. Disconnect Between Generative and Forecasting Objectives

The paper's central premise is that a "generative" VAE-based architecture is superior for forecasting transfer learning. However, the connection between these two goals is weak and poorly supported by the experiments.

Conflicting Goals: The generative goal (reconstructing the past, $X_t$) and the forecasting goal (predicting the future, $Y_{t+1:t+h}$) are not inherently aligned. Optimizing for a perfect, high-fidelity reconstruction can force the model to learn noisy, high-frequency details from the input that are useless or even detrimental to forecasting.

Missing Analysis: The paper provides no analysis to show these two objectives are synergistic. The authors should have included:

A qualitative analysis showing the model's reconstructions ($\hat{X}_t$).

A visualization of the latent space ($z$) to show whether it clusters semantically similar time series (e.g., by domain or seasonality), which would support the claim that it captures "high-level representations."

Without this, the "generative" component feels like an unproven and potentially unnecessary complication.

2. Unclear Source of Zero-Shot Generalization

The paper claims SOTA zero-shot performance but fails to convincingly attribute this capability to its novel architecture. The skepticism that a simple MLP-based model can "naturally" achieve this level of generalization is warranted because the paper fails to provide the necessary evidence.

Insufficient Ablation: The ablation study (Table 2) is too limited. It shows that removing components hurts performance, but it doesn't isolate the key ingredient. A crucial missing experiment is comparing TimeGEN to a non-variational counterpart (i.e., a standard autoencoder with only the $\text{Recon}(\cdot)$ loss, no $\text{KL}(\cdot)$ term).

Unanswered Questions: It is unclear if the success comes from the VAE's latent $z$, the joint loss, the specific decoder, or simply the temporal normalization. The paper does not provide the analysis needed to distinguish between these factors, making the core architectural claims unsubstantiated.

3. Lack of Detail on Variational Training Dynamics

The paper introduces a Variational Autoencoder, including a $\beta \cdot \text{KL}(q_{\phi}(z|X_t) || \mathcal{N}(0,I))$ term in its loss function, but completely glosses over the significant challenges associated with training such a model.

Training Instability: VAEs are notoriously difficult to train. They can suffer from "posterior collapse" (where the KL term goes to zero and the latent space is ignored) or require careful balancing of the reconstruction, forecasting, and KL loss terms.

Missing Details: The paper mentions $\beta$ is "linearly annealed" (Section 3.3) but provides no further details. This is a critical hyperparameter. A robust analysis would require details on the annealing schedule, the final value of $\beta$, and the sensitivity of the model's performance to this parameter. This omission is a significant gap in reproducibility and a major unaddressed technical challenge.

**Questions:**

See the weakness.

---

> ### Author Response · Authors · 2025-11-22
>
> We thank the reviewer for their thoughtful and constructive review, specifically the recognition of our contributions as well as the extension and breadth of our experimental evaluation.
>
> Regarding the weaknesses and concerns raised by the reviewer, we are confident we can address them as follows.
>
> 1. Our perspective is that the two terms are complementary: the reconstruction term aligns forecasts with historical dynamics by encouraging latent representations that capture underlying structure, while the forecasting term promotes generalization across time. To clarify this, we expanded our ablations by (1) removing the reconstruction loss and (2) replacing the variational autoencoder with a non-generative autoencoder. The results show that although all variants behave similarly in full-shot settings, both the reconstruction loss and the variational mechanism are essential for strong cross-domain performance. Removing reconstruction greatly reduces transfer accuracy, which conveys its central role.
>
> |Variant|Full-shot (single-domain)|In-domain (single-domain)|Single-source (cross-domain)|
> |-|-|-|-|
> |**TimeGEN**|1.33|1.41|**1.93**|
> |**Non-Generative Encoder**|1.35|1.32|2.24|
> |**No Reconstruction**|1.36|1.46|9.85|
>
>    We thank the reviewer for their valuable suggestions (e.g., visualization of reconstructions and the latent space) and agree that these are promising directions we will explore in future work. We prioritized an extensive ablation study to directly assess the impact of these model components.
>
> 2. We have substantially revised the methodological section to clarify the design motivations and added new ablations to address the missing comparisons. The results below show that the zero-shot generalization capability does not originate from a single component but from the interaction of several complementary ones. We hope that the updated explanation and new ablation experiments clarify this point and address the reviewer’s concerns.
>
>    We now explicitly explain why each component was selected:
>
>    - Our variational encoder encourages smooth, domain-general latent representations instead of memorizing domain-specific patterns. Mapping inputs to a distribution regularizes the latent space toward stable temporal structure that transfers across domains. The “Non-Generative Encoder” variant confirms the weaker cross-domain performance.
>
>    - In the joint reconstruction and forecasting objectives, the reconstruction captures local, fine-grained patterns, while forecasting enforces long-range temporal dependencies. Removing reconstruction (“No Reconstruction”) substantially degrades cross-domain accuracy.
>
>    - Temporal normalization addresses scale differences and noise across domains. Removing it (“No Normalization”) produces large errors in the cross-domain transfer.
>
>    - MLPs keep the architecture lightweight while providing a flexible mapping to the latent space. Decoder components (basis expansion + multi-rate interpolation) model high-frequency and long-range structure. When we restrict them (“Shallow Latent,” “Shallow + No Mixture”), we see a drop in cross-domain accuracy.
>
>    - Injecting the latent representation into each decoder block ensures global information influences the entire forecast, stabilizing training and increasing expressiveness of the latent space. Supplementary material now includes additional details and empirical evidence.
>
> |Variant|Full-shot (single-domain)|In-domain (single-domain)|Single-source (cross-domain)|
> |-|-|-|-|
> |**TimeGEN**|1.33|1.41|**1.93**|
> |**Non-Generative Encoder**|1.35|1.32|2.24|
> |**No Reconstruction**|1.36|1.46|9.85|
> |**No Normalization**|1.32|1.33|5.97|
> |**Shallow Latent**|1.31|1.32|2.50|
> |**Shallow + No Mixture**|1.33|1.33|2.72|
>
> 3. We acknowledge that the original paper lacked detail on VAE training, especially the annealing process. We summarize these details here and in the revised version of the paper. We use linear KL annealing: \beta(t) ramps from 0 to 0.3 over 500 steps to prevent early posterior collapse, stabilize gradients, and allow the model to first learn reconstruction/forecasting. We also use other mechanisms to avoid collapse and numerical issues, namely clamping of log-variance in both KL and reparameterization.
>
>    Although collapse can still occur with an expressive decoder, two architectural choices reduce its likelihood: (1) explicit latent injection into decoder blocks, creating a direct dependency on the latent variables, and (2) combining reconstruction and forecasting, which forces the latent space to encode complementary information. We will include empirical evidence for these effects in the final version.
>
>    All code is available for full reproducibility. While we could not run an exhaustive sensitivity analysis within the rebuttal timeline, we tested \beta values in [0.1, 0.5] and observed stable training and comparable performance. A systematic sensitivity study will be included in the revised version.

---

### Official Review · Reviewer_WfdP · 2025-11-01

**Soundness:** 2
**Presentation:** 2
**Contribution:** 2
**Rating:** 4
**Confidence:** 3

**Summary:**

This paper proposes TimeGEN, an efficient temporal generative encoder network focusing on transfer learning for time series forecasting. Using a variational encoder, it extracts global latent temporal features shared across different series and domains. Its modular decoder architecture captures multiscale dynamics via neural basis expansion and hierarchical interpolation. The authors extensively test TimeGEN across standard and transfer learning scenarios on heterogeneous public datasets. Results show good generalization, particularly in cross-domain and zero-shot tasks, with substantial speedups in training compared to Transformer and other MLP-based approaches.

**Strengths:**

1. TimeGEN reduces zero-shot and cross-domain forecasting errors compared to state-of-the-art methods.
2. Achieves speedup in training time relative to Transformer and other MLP-based models, making it highly scalable.
3. Uses a variational encoder with a joint reconstruction and forecasting loss to shape a latent space capturing both local and global temporal dependencies.
4. Combines neural basis expansion for high-frequency patterns and multi-rate interpolation for long-term trends, balancing local detail and global structure.

**Weaknesses:**

1. The author mentioned GENERATIVE in their title, but what specific generative way should be in their method? The author should clarify it.
2. Since it is claimed as a generative model, and the LLM is an unignorable technical in GenAI. It is suggested to compare their method with some LLM-based time series forecasting methods.
3. I am concerned about the necessity of Figure 2.
4. The framework is defined as a lightweight structure. How to prove this claim?
5. I am concerned about the decoder part. Why does the author introduce two blocks? Will it improve the complexity?

**Questions:**

See above

---

> ### Author Response · Authors · 2025-11-22
>
> We appreciate the reviewer’s comments and are confident we can address them as follows.
>
> 1. We employ a variational autoencoder as the generative component of our model, enabling TimeGEN to learn a probabilistic latent representation from input time series data. This generative approach allows the model to sample from the learned latent space and capture the underlying data distribution across heterogeneous time series datasets, in both input reconstruction and forecasting.
>
>    To assess the impact of this generative VAE component, we have extended our ablation analysis to include a variant where we replace the VAE with a standard (non-generative) autoencoder and another variant that removes the reconstruction loss term. This comparison helps clarify the role and benefit of the generative mechanism in our architecture. The results are as follows:
>
>    | Variant  | Full-shot (single-domain) | In-domain (single-domain) | Single-source (cross-domain) |
>    |--|--|--|--|
>    | **TimeGEN** | 1.33|1.41| **1.93**|
>    | **Non-Generative Encoder**| 1.35 | 1.32  | 2.24 |
>    | **No Reconstruction** |1.36 |1.46  |9.85 |
>
>    The results indicate that, while the impact of the variational and reconstruction components in full-shot and in-domain settings is generally small, they are key ingredients in the cross-domain transfer learning capabilities of TimeGEN.
>
> 2. We do not include LLM-based forecasting methods in the experiments for two main reasons. First, recent evidence (e.g., a NeurIPS’24 paper [1], see abstract) indicates that simpler attention layers, such as those used in the Transformer-based baselines we already evaluate, outperform LLM-specific architectures for time series forecasting. Second, while we do not directly include LLM-based models, our comparisons span a broad set of strong Transformer-based approaches, such as TimeMOE (ICLR’25 Spotlight), PatchTST, iTransformer, and TFT. Our selection of baselines is consistent with prior top-conference benchmarks (e.g., ICLR and NeurIPS [2,3]), which also do not evaluate LLM-based models in this domain.
>
>    Beyond the improved ablation study described above, we have further expanded our experiments to include xLSTM (NeurIPS’24), TimeMixer (ICLR’24), and NBEATS (ICLR’20) to improve the experimental evaluation. Experimental results continue to support the effectiveness and generalizability of our proposed approach relative to these strong baselines.
>
>    | Method | MASE ( multi-source cross-domain) |
>    |--|--|
>    | **TimeGEN** | **1.493** |
>    | **NBEATS**  | 2.096 |
>    | **TimeMixer** | 1.888  |
>    | **xLSTM** | 1.725    |
>
> 3. Our evaluation procedure defines four complementary evaluation regimes covering full data availability, generalization to similar distributions, and out-of-domain transfer, both from single and multiple sources. Figure 2 provides a visual, intuitive overview of the relationships among these regimes.
>
> 4. We define the framework as lightweight because it is fully MLP-based, without attention, recurrent, or convolutional layers. While we do not explicitly count parameters or provide a formal complexity analysis, we demonstrate its efficiency empirically: as shown in Table 1, the proposed method achieves substantially faster training times compared to other approaches, supporting our claim of a lightweight design. All experiments were executed under similar methodological (e.g., hyperparameter tuning) and computational conditions to ensure fairness.
>
> 5. We use two types of blocks, neural hierarchical interpolation (NHI) blocks and neural basis expansion (NBE) blocks, to capture the diversity of temporal patterns in time series data. NHI blocks operate on downsampled versions of the time series, modeling long-term behavior, while NBE blocks use flexible learned basis expansions to capture finer-grained details and short-term fluctuations.
>
>    Our working hypothesis is that interleaving these block types enables the decoder to simultaneously model both global and local dynamics, leading to better performance. The number and order of blocks are determined via hyperparameter search for each dataset. This hypothesis is supported by the ablation study, which shows that using a shallow latent space and restricting the decoder to a single block type produces a noticeable drop in cross-domain accuracy.
>
>    | Variant   | Full-shot (single-domain) | In-domain (single-domain) | Single-source (cross-domain) |
>    |--|--|--|--|
>    | **TimeGEN** | 1.33 | 1.41 | **1.93** |
>    | **Shallow Latent** | 1.31 | 1.32 | 2.50 |
>    | **Shallow + No Mixture** | 1.33 | 1.33 | 2.72 |
>
> [1] Tan, Mingtian, et al. *Are language models actually useful for time series forecasting?* Advances in Neural Information Processing Systems 37 (2024): 60162–60191.
> [2] Wang, Shiyu, et al. *TimeMixer: Decomposable Multiscale Mixing for Time Series Forecasting.* ICLR (2024).
> [3] Shi, Xiaoming, et al. *Time-MoE: Billion-Scale Time Series Foundation Models with Mixture of Experts.* ICLR (2025).

---

> > ### Comment · Reviewer_WfdP · 2025-11-27
> >
> > Thanks for the authors' responses and I decide to keep my score.

---

### Official Review · Reviewer_FqVZ · 2025-11-01

**Soundness:** 3
**Presentation:** 3
**Contribution:** 2
**Rating:** 4
**Confidence:** 4

**Summary:**

The paper introduces TimeGEN, a lightweight, MLP-based generative deep learning architecture for time series forecasting, specifically aimed at addressing challenges in cross-domain transfer learning. TimeGEN uses a variational encoder to capture high-level temporal representations across diverse time series domains, and employs a modular decoder architecture for multiscale forecasting. The model integrates reconstruction and forecasting loss functions, along with temporal normalization to ensure robustness against varying scales and noise. Empirical evaluations across ten public datasets demonstrate that TimeGEN outperforms state-of-the-art (SOTA) methods in zero-shot and cross-domain settings, achieving significant improvements in forecasting accuracy and computational efficiency.

**Strengths:**

Strengths:
1. The studied problem is interesting and important for AI area.

2. The proposed TimeGEN is significantly more efficient in terms of training time, offering a 2–30× speedup compared to more complex models like Transformers.

**Weaknesses:**

Weaknesses:

1. There is a lack of comparison with the latest relevant literature. The most recent methods compared in this paper are TimeMOE (ICLR 2025) and KAN (ICLR 2025). However, TimeMOE is a method for large-scale basic time series models, and its core achievement is in large-scale, efficient, and general time series prediction, rather than specifically for cross-domain time series prediction, which is the focus of this paper. KAN networks, on the other hand, focus on more general neural network models, rather than time series problems. Other comparison methods are limited to those before 2023, lacking comparison with the latest and most relevant results, such as Unitime [s1] and LPTM [s2].

2. While the empirical results are good, the paper does not provide a deep theoretical explanation for some of the architectural decisions, such as the combination of reconstruction and forecasting loss. There is limited discussion on why certain components, such as temporal normalization or the specific use of MLP-based models, are preferable over other design choices.

3. While the ablation study presents useful insights into the model's components, the exploration could be expanded further. The removal of certain blocks (e.g., temporal normalization or the deep latent conditioning) leads to large performance drops, yet the precise role of each component in the architecture is not fully unpacked. More granular analysis of how each component contributes to the overall performance, especially in cross-domain settings, would clarify the model’s design rationale.


[s1] Liu, Xu, et al. "Unitime: A language-empowered unified model for cross-domain time series forecasting." Proceedings of the ACM Web Conference 2024. 2024.

[s2] Prabhakar Kamarthi, Harshavardhan, and B. Aditya Prakash. "Large Pre-trained time series models for cross-domain Time series analysis tasks." Advances in Neural Information Processing Systems 37 (2024): 56190-56214.

**Questions:**

You report 2–30× faster training times (Table 1). Are these gains primarily due to architectural simplicity (MLP-only) or optimization choices (e.g., batch size, windowing, or fewer parameters)?

---

> ### Author Response · Authors · 2025-11-22
>
> We appreciate the reviewer’s comments and are confident we can address them as follows.
>
> 1. Following your suggestion, we have substantially improved our experimental baselines. In the updated manuscript, we include additional recent models: TimeMixer (ICLR’24), xLSTM (NeurIPS’24), and NBEATS (ICLR’20). Regarding Unitime [s1] and LPTM [s2], we acknowledge their importance and novelty. However, we did not include them because their codebases are not directly compatible with *neuralforecast*, the forecasting framework we use. Therefore, we could not ensure a faithful implementation in time for the rebuttal. Nevertheless, we commit to including these two approaches in a revised version.
>
> | Method | MASE ( multi-source cross-domain) |
> |--|--|
> | **TimeGEN**   | **1.493** |
> | **NBEATS**    | 2.096  |
> | **TimeMixer** | 1.888  |
> | **xLSTM**     | 1.725 |
>
>
> 2. and 3. We revised the methodological section to enhance the motivation and provide a clearer, more detailed explanation of each architectural component and how they contribute to the overall behavior. For clarity, we also summarize these explanations below.
>
>    Our work is primarily empirical, and the goal of the paper is to identify a simple, scalable architecture that performs well across heterogeneous time-series domains. While we do not provide a formal theoretical analysis for each architectural choice, our design decisions were grounded in expected contributions of each component. These motivations were not clearly articulated in the submitted version, but we can do so in the revision, as follows:
>
>
>    - **Variational encoder**: We use a variational encoder because it encourages smooth and domain-general latent representations rather than memorizing domain-specific patterns. By mapping each input to a distribution and keeping it close to the prior, the stochastic latent space regularizes the model and forces it to encode only the stable temporal structure that transfers across domains.
>      To support this, we added a new variant to our ablation study: replacing the VAE with a simpler autoencoder. This variant (“Non-Generative Encoder”) produces weaker cross-domain performance, as shown in the table below.
>
>    - **Joint reconstruction and forecasting objectives**: Reconstruction and forecasting require different information from the latent space. Reconstruction encourages the model to capture local, fine-grained patterns of the input, whereas forecasting enforces representation of long-range temporal dependencies. In our ablations, removing the reconstruction loss (“No Reconstruction”) results in a significant degradation in cross-domain accuracy.
>
>    - **Temporal normalization**: Temporal normalization is included to handle scale differences and noise characteristics that arise across domains. The “No Normalization” variant yields large errors in the single-source regime, demonstrating its importance for stable transfer.
>
>    - **MLP**: We adopt MLPs to keep the architecture lightweight. An MLP provides a simple and flexible mapping to the latent space, which we want to be expressive. We use neural basis expansion modules for high-frequency details and multi-rate interpolation blocks for long-range trends. The “Shallow + No Mixture” and “Shallow Latent” variants show that reducing decoder flexibility or weakening the latent conditioning consistently harms cross-domain accuracy.
>
>    - **Latent conditioning of decoder blocks**: We condition each decoder block directly on the latent representation to ensure that the global information captured in the latent space influences the full forecasting process. This stabilizes training and improves representational expressiveness. We have now expanded the supplementary material with these explicit details and empirical evidence.
>
> | Variant | Full-shot (single-domain) | In-domain (single-domain) | Single-source (cross-domain) |
> |--|--|--|--|
> | **TimeGEN**   | 1.33  | 1.41      | **1.93**       |
> | **Non-Generative Encoder** | 1.35 | 1.32         | 2.24   |
> | **No Reconstruction**      | 1.36  | 1.46        | 9.85    |
> | **No Normalization**       | 1.32  | 1.33       | 5.97   |
> | **Shallow Latent**         | 1.31      | 1.32         | 2.50  |
> | **Shallow + No Mixture**   | 1.33     | 1.33  | 2.72   |
>
>
>
> Regarding the question about training efficiency, we note that, to ensure a fair comparison, all methods were independently tuned using random search. This process optimized key parameters such as batch size and input size, among other training hyperparameters, individually for each method as defined in the appendix of the paper. Therefore, we believe the substantial training speedups are primarily driven by the architectural simplicity of our MLP-based design.

---

### Official Review · Reviewer_6yvs · 2025-11-03

**Soundness:** 3
**Presentation:** 2
**Contribution:** 3
**Rating:** 4
**Confidence:** 4

**Summary:**

The paper proposes a lightweight, MLP-based generative framework named TimeGEN for cross-domain time series forecasting. It employs a variational encoder for transferable high-level temporal representations, a modular decoder for long-range trends with high-frequency variations, temporal normalization for varying input and trains the model on objective that combines reconstruction, forecasting, and variational regularization to enhance generalization. Experiments on ten datasets demonstrate superior zero-shot and cross-domain generalization with faster training speed.

**Strengths:**

1.	TimeGEN adopts MLP-based architecture rather than a Transformer, contributing to the growing line of MLP-based foundation models for time series forecasting. This design choice offers a promising and efficient alternative to attention-based approaches.
2.	The model design is computationally efficient and demonstrates strong empirical performance when compared against seven diverse baseline methods, indicating its effectiveness across diverse architectures.
3.	The experimental setup is comprehensive, covering both full-shot and three zero-shot transfer settings. The inclusion of ten datasets provides sufficient evidence of the model’s robustness and generalization ability across diverse domains.

**Weaknesses:**

1.	The proposed model adopts a relatively simple architecture with an MLP-based encoder. While the authors claim that it can capture high-level temporal representations across diverse domains, it remains unclear how such a simple structure is capable of modeling complex dependencies and non-stationary patterns that typically arise in heterogeneous time series scenarios. A more detailed explanation or empirical evidence would strengthen this claim.
2.	Although the baseline selection covers multiple architecture families, the overall comparison remains limited — only seven baselines are included, among which there is only one MLP-based model and one state-of-the-art foundation model. This makes the evaluation less convincing. The authors are encouraged to include more recent and relevant baselines, especially MLP-based architectures and stronger foundation models, to ensure a fair and comprehensive comparison.
3.	As reported in Table 1, the proposed model achieves the fastest training speed (normalized = 1.0) compared with PatchTST (≈ 5× slower). This likely implies that the model has substantially fewer parameters. However, it raises the question: how can a small-capacity model effectively encode and transfer diverse pre-trained knowledge across domains? Moreover, both TimeGEN and TSMixer are MLP-based, yet the reported training time of TSMixer (31.042× slower) appears inconsistent. The authors should clarify this discrepancy and explain the source of such a large gap.
4.	The paper emphasizes temporal normalization and the joint reconstruction–forecasting loss as key components. However, both techniques are common practices in recent time series models and should not be presented as methodological contributions. The authors are advised to reposition these components as supporting techniques rather than major novelties.

**Questions:**

Please check above section.

---

> ### Author Response · Authors · 2025-11-22
>
> We appreciate the reviewer’s comments and are confident we can address them as follows.
>
> 1. Our working hypothesis is that an MLP-based VAE is capable of effectively modeling the heterogeneous patterns typically present in time series datasets. To provide further empirical evidence supporting this hypothesis, we have extended our ablation study by including a variant in which the VAE is replaced with a non-generative autoencoder. This allows us to directly assess the impact of the generative component.
>
>    As shown in the table below, replacing the variational encoder with a standard autoencoder weakens our transfer performance. In addition, the previous version of the ablation study already demonstrated that a key factor behind the effectiveness of the encoder in cross-domain settings is the direct conditioning of each decoder block on the latent space. When this connection is removed in the “Shallow Latent” variant, cross-domain performance drops significantly.
>
> | Variant                   | Full-shot (single-domain) | In-domain (single-domain) | Single-source (cross-domain) |
> |--|--|--|--|
> | **TimeGEN**               | 1.33                       | 1.41                       | **1.93**                      |
> | **Shallow Latent**        | 1.31                       | 1.32                       | 2.50                          |
> | **Non-Generative Encoder**| 1.35                       | 1.32                       | 2.24                          |
>
> 2. We added three new architectures to our experimental comparison: two additional MLP-based models (NBEATS and TimeMixer) and xLSTM. TimeMixer has been published in ICLR’24 and xLSTM has been published in NeurIPS’24. By including these three approaches, we strengthen our claims about generalization and efficiency of TimeGen, and provide a more comprehensive evaluation against additional recent approaches.
>
> | Method | MASE ( multi-source cross-domain) |
> |--|--|
> | **TimeGEN**   | **1.493**                        |
> | **NBEATS**    | 2.096                            |
> | **TimeMixer** | 1.888                            |
> | **xLSTM** |  1.725                           |
>
>    Concerning foundation models, we include three architectures that are the backbone of foundation models, specifically: TimeMOE, which was published in ICLR’25 as a spotlight paper;  TSMixer, which is the core architecture for the TTM model [2]; and xLSTM, which is the architecture behind the TiRex foundation model [3]. We focus on evaluating architectures instead of pre-trained models since this provides more flexibility in terms of the evaluation procedure. This is because most foundation models do not typically disclose the training data, and that prevents us from ensuring no data leakage occurs.
>
> 3. Our findings about the computational complexity of TSMixer are consistent with the computational costs reported in the original paper proposing TSMixer [1, see Table 6]. For example, they report that TSMixer is 4.6× slower than PatchTST, which aligns with our findings. Since we observed PatchTST to be more than 5× slower than TimeGEN, this is how we arrive at the empirical evidence that TSMixer is ~30× slower than TimeGEN. We have made this clearer in the revised manuscript and added an explicit mention of these prior results.
>
> 4. We agree that temporal normalization and composite loss functions are not novel. The novelty stems from how these are integrated in a unified generative framework using a variational encoder alongside a modular decoder. The synergy of these components enables improved cross-domain transfer, and better generalization. It produces a latent space that is robust to scale and noise variations (via normalization), captures the underlying temporal structure (via reconstruction), and remains aligned with future dynamics (via forecasting).
>
>     We have improved the motivation for using these components and expanded our ablation experiments by introducing a new variant that removes the reconstruction loss component.  We found that the reconstruction loss term is critical for its competitive cross-domain forecasting accuracy. The previous version of the ablation study already showed the importance of temporal normalization for the cross-domain transfer.
>
> | Variant                 | Full-shot (single-domain) | In-domain (single-domain) | Single-source (cross-domain) |
> |--|--|--|--|
> | **TimeGEN**            | 1.33      | 1.41      | **1.93**       |
> | **No Normalization**   | 1.32      | 1.33      | 5.97           |
> | **No Reconstruction**  | 1.36      | 1.46      | 9.85           |
>
> [1] Chen et al. TSMixer: An All-MLP Architecture for Time Series Forecasting. TMLR, 2023.
>
> [2] Ekambaram et al. Tiny Time Mixers (TTMs). NeurIPS 2024.
>
> [3] Auer et al. TiRex: Zero-Shot Forecasting Across Long and Short Horizons. ICML FMSD Workshop, 2023.

---

### Meta-Review · Area_Chair_nHkJ · 2026-01-02

**Summary:**

This paper proposes TimeGEN, a lightweight MLP-based generative architecture for cross-domain and zero-shot time series forecasting. The method combines a variational encoder, a modular multi-scale decoder, and a joint reconstruction-forecasting objective. The overall design philosophy is appealing. The evaluation protocol is good, covering multiple regimes and heterogeneous datasets, and the results demonstrated strong efficiency advantages compared to Transformer-based models.

Despite these strengths, all reviewers raised concerns regarding the clarity of presentation, the completeness of the analysis, and the substantiation of several core claims. In particular, the paper’s storytelling and articulation of technical motivations are below average, and key aspects of the generative formulation and transfer mechanisms are insufficiently justified. While the rebuttal addresses several reviewer comments and adds additional ablation results, the overall work remains less complete in its current form and would benefit from substantial polishing and deeper analysis before being ready for acceptance. Additional side analyses and improved storytelling would be required to convincingly support the central claims.

**Reviewer Concerns:**

Reviewers 6yvs, WfdP, and FqVZ raised consistent concerns about clarity, storytelling, and justification of key claims (e.g., what the "generative" component contributes to forecasting transfer, why a lightweight MLP/VAE can encode transferable knowledge, and how the decoder components and normalization drive generalization). Reviewer FqVZ additionally emphasized missing comparisons with the latest cross-domain forecasting models and limited depth in the methodological motivation. Reviewer CsL5 raised more fundamental issues regarding the alignment between reconstruction and forecasting objectives, unclear attribution of zero-shot generalization, and insufficient detail on VAE training dynamics.

The rebuttal addresses several of these points by adding ablations, clarifying training efficiency, expanding baselines, and providing additional implementation details. However, several concerns are only partially resolved (e.g., those raised by CsL5), and key issues around presentation quality, completeness of analysis, and strength of attribution remain.

**Reviewer Scores:**

Reviewers 6yvs, WfdP, and FqVZ would likely maintain their marginally below threshold scores, with at most minor softening due to the additional experiments and clarifications. Reviewer WfdP explicitly indicated no score change after the rebuttal. Reviewer CsL5 would likely remain negative, as their core concerns about generative justification and analytical completeness were only partially addressed. Overall, the aggregate assessment would remain below the acceptance threshold.

---

### Decision · Program_Chairs · 2026-01-26

Reject